# The Potential of *Esteya* spp. for the Biocontrol of the Pinewood Nematode, *Bursaphelenchus xylophilus*

**DOI:** 10.3390/microorganisms10010168

**Published:** 2022-01-13

**Authors:** David Pires, Cláudia S. L. Vicente, Maria L. Inácio, Manuel Mota

**Affiliations:** 1Instituto Nacional de Investigação Agrária e Veterinária (INIAV, I.P.), 2780-159 Oeiras, Portugal; david.pires@iniav.pt; 2Mediterranean Institute for Agriculture, Environment and Development (MED), University of Évora, Pólo da Mitra, Apartado 94, 7006-554 Evora, Portugal; mmota@uevora.pt; 3GREEN-IT Bioresources for Sustainability, ITQB NOVA, Av. da República, 2780-157 Oeiras, Portugal

**Keywords:** *Bursaphelenchus xylophilus*, pine wilt disease, nematophagous fungi, biological control, *Esteya*

## Abstract

The pinewood nematode (PWN), *Bursaphelenchus xylophilus*, is the causal agent of pine wilt disease (PWD) and a quarantine organism in many countries. Managing PWD involves strict regulations and heavy contingency plans, and present climate change scenarios predict a spread of the disease. The urgent need for sustainable management strategies has led to an increasing interest in promising biocontrol agents capable of suppressing the PWN, like endoparasitic nematophagous fungi of the *Esteya* genus. Here, we review different aspects of the biology and ecology of these nematophagous fungi and provide future prospects.

## 1. Introduction

The pinewood nematode (PWN), *Bursaphelenchus xylophilus* (Steiner and Buhrer, 1934) Nickle, 1970, is the causal agent of pine wilt disease (PWD) and a quarantine organism in many countries [1,2,3]. It is thought to be native to North America, where autochthonous pine species show some degree of tolerance, in contrast to exotic species that are usually susceptible to the nematode. The PWN is considered the sixth most economically important plant-parasitic nematode [4] and a major threat to pine forests worldwide, especially in Asia and Europe. *Bursaphelenchus xylophilus* is carried and transmitted by pine sawyer beetles belonging to the *Monochamus* genus [5,6]. There are currently 13 species of *Monochamus* known to be vectors of the PWN worldwide [7,8]. Due to its economic importance and worldwide dissemination, an enormous effort has been devoted to understand which factors play a determinant role in the epidemiology of PWD [9]. The disease results from complex interactions between the nematode, its insect vector and host plants (mostly *Pinus* spp.), the nematode being the common element in this interaction [10,11,12]. Management of PWD involves strict regulations and heavy contingency plans, resulting in the felling, removal and destruction of infected trees, with limited effectiveness [13], having a serious economic impact on the exploration and commercialization of timber and other wood products [14,15,16]. Aside from the economic setback, PWD threatens biodiversity, disrupts forest ecosystems and hampers resilience to climate change [17,18]. In fact, current scenarios strongly suggest that increasing temperatures and climate change are likely going to result in a prevalence of the PWN and spread of PWD to areas free of the disease [19,20,21]. Consequently, the urgent need for sustainable management strategies has led to an increasing interest in organisms capable of suppressing the PWN [22,23,24]. The endoparasitic nematophagous fungi of the *Esteya* genus are reported as natural enemies of the PWN and promising biocontrol agents. There are currently two described species: *E. vermicola* [25] and *E. floridanum* [26]. These PWN antagonists have received great interest as biocontrol agents [27]. *Esteya vermicola* was successful in suppressing the PWN both in greenhouse experiments [28] and field assays [29]. On the other hand, *E. floridanum* was able to kill PWNs in vitro but only delayed the death of trees in vivo [26], so more research is needed to assess its potential in controlling the PWN. Since its discovery, *E. vermicola* has been patented in the United States and South Korea as a promising option for the management of PWD [25,30,31]. Here, we review current research on the taxonomy, phylogeny, morphology, distribution and host range of *Esteya* spp., molecular mechanisms behind their parasitism, application of *Esteya* spp. to improve PWD management and future perspectives.

## 2. Taxonomy, Morphology and Distribution of *Esteya* Species

*Esteya* is a ubiquitous fungal genus belonging to the Ophiostomatales order (Ascomycota, Sordariomycetes), with only two species currently described, *E. vermicola* [25] and *E. floridanum* [26], isolated from different substrates around the world. As members of the Ophiostomataceae family, mitochondrial genome comparisons show that *Esteya* spp. are closely-related to *Sporothrix* [32] and share similar characteristics to other fungi of the same family: a saprophytic lifestyle and endophytic to weakly phytopathogenic. However, some species have been reported to kill healthy trees when the inoculum is too high [33].

### 2.1. Taxonomy and Phylogeny

The first isolate of *E. vermicola* was deposited as *Lunuromyces thunbergii* [34,35], until a new genus was created and the fungus described in 1999 [25] (Table 1).

Molecular tools are crucial for the accurate identification and characterization of existing isolates and even new species. The genomic DNA of *E. vermicola* was successfully extracted from mycelia obtained from agar plates, liquid media and directly from pine trees. Wang et al. [36] described a simple method to extract DNA from mycelia grown in potato dextrose broth (PDB) using cetyltrimethylammonium bromide (CTAB) and later outlined a protocol for DNA extraction from agar plates [37]. A simple and effective sample preparation method was optimized by Yin et al. [38] for the DNA extraction of *E. vermicola* from pine trees. Briefly, it consists in dipping samples in sterile water overnight, with constant shaking, prior to using a commercially available DNA extraction kit. When this step was overlooked, performing a standard protocol did not yield enough fungal DNA to be detected by polymerase chain reaction (PCR). Genomic DNA from *E. floridanum* was extracted using an Extract-N-Amp Plant PCR kit with modifications [26]. Many molecular markers and their respective primer combinations have been proposed to identify *Esteya* to species level, but the most used is β-tubulin (βT) (Bt2a/Bt2b) [39,40]. Other markers include the translation elongation factor 1-α (EF1-α) (EF1-983F and EF1-2218R) [41], the nuclear large subunit (LSU) ribosomal DNA (rDNA) (LR0R/LR5) [42,43] and the transcribed spacer regions (ITS1 and ITS2) ITS1-5.8S-ITS2 rDNA locus (ITS1F/ITS4) [44,45]. Other authors noted that ITS and LSU markers did not clearly distinguish closely-related species within the Ophiostomatales order, but were useful in assigning isolates to the species complexes or unknown lineages within *Ophiostoma s. l.*, *Leptographium s. l.*, and *Sporothrix*, whereas βT and TEF1-α were robust enough to identify isolates to the species level [46]. There are currently 35 nucleotide sequences available in the National Center for Biotechnology Information taxonomy database: 27 belonging to *E. vermicola*, 3 to *E. floridanum* and 5 to unclassified *Esteya* spp. [47].

Phylogenetic data have shown that *Esteya*, specifically *E. vermicola*, diverged from plant-associated *Grosmannia clavigera* about 135 million years ago [50], and other phylogenetic placements based on LSU sequences revealed that *E. vermicola* evolved independently to other Ophiostomatales, being closely related to the ambrosia beetle mutualist *Raffaelea sulphurea* [51]. There are similarities between the morphology of *Esteya* and *Leptographium*, but the nematophagous ecology of the former and the morphology of its infectious conidia are very different from other species of *Leptographium s. l.* Thus, it is clear that *Esteya* belongs in the Ophiostomatales order, but the taxonomy of this order has undergone considerable revisions in recent years and there are groupings for which monophyly is not certain, such as *Leptographium s. l.* (which includes *Grosmannia* species, the *R. sulphurea* species complex, and *E. vermicola*) [52,53]. Consequently, the status of the genus *Esteya* is still unresolved [52,54].

Phylogenetic analyses based on the peptide sequences of serine proteases from various entomopathogenic and nematophagous fungi revealed a closer evolutionary relationship between *E. vermicola* NKF 13222 and the toxin-producing fungus *Clonostachys rosea*, rather than with the trapping fungus *Dactylellina haptotyla* [55]. Further evolutionary relationships between *E. vermicola* strains, CBS 100821, ATCC 74485, CBS 115803 and CNU 120806, placed them in a well-supported monophyletic clade, closely related to *Pseudomonas stutzeri* and affiliated with *Gammaproteobacteria* [56]. Whole mitochondrial genome comparisons showed that *E. vermicola* CBS 115803 clustered together with the genus *Sporothrix* [32]. Interestingly, phylogenetic relationships between *E. vermicola* from different geographical origins showed that European isolates are in the same sub-clade, whereas the Brazilian, Korean and Taiwanese isolates are in a different sub-clade, although belonging to the same clade [49]. In relation to other *Esteya* species, Li et al. [26] showed that *E. floridanum* is phylogenetically closer to the Chinese isolate CXY 1893 of *E. vermicola*.

### 2.2. Morphology

Macroscopically, *E. vermicola* colonies (Figure 1a) appear greyish-green to dark-green and reverse dark-grey to olive-green after being cultured for 7 days on rich media [25,37,48]. Other authors have reported slight variations: grayish-blue with slight white on the surface and reverse dark-grey to chartreuse [35], white at first, gradually turning grayish-green and dark-green with dark-brown pigments [36,49]. In similar conditions, *E. floridanum* colonies (Figure 1b) appear initially white, turning light-grey after 5 days, and reverse initially white, switching to dark-brown on potato dextrose agar (PDA) and beige on malt extract agar (MEA) [26].

Both *Esteya* spp. are microscopically very similar. *Esteya vermicola*’s features include subhyaline hyphae, grey-green to olive-green, smooth to roughened, sometimes with a slime sheath. The fungus produces two types of conidiophores, conidiogenous cells and conidia, lunate and bacilloid [25,48]. Conidiogenous cells of the first type are sessile, smooth to roughened, with an olive-green inflated base, abruptly tapering into a thin subhyaline neck of varying length, often crooked and sometimes percurrent. Conidia are one-celled, lunate, subhyaline and smooth-walled, but their characteristics can vary between isolates [25,35,36,37,48]. For instance, ATCC 74485 and FXY121 lunate conidia are solitary, while CBS 115803, CNU 120806 and CBS 100821 can produce 1–4 spores by the same conidiogenous cell [35,48]. Conidiophores of the second type are loosely branched or simple, olive-green, often roughened, conidiogenous cells straight, mostly with a swollen base. Conidia cylindrical or bacilloid, one-celled, hyaline and smooth [25,48]. On the other hand, *E. floridanum* seems to only produce solitary lunate and bacilloid conidia, and both types of spores were observed to attach to PWNs [26]. While conidiogenous cells of *E. vermicola* are formed directly on the main vegetative hypae, conidiogenous cells of *E. floridanum* are often supported by a short, swollen cell, which connects to the main hyphae. The teleomorph of *Esteya* spp. was never observed.

### 2.3. Distribution and Host Range

*Esteya vermicola* was first isolated in Japan [35] and described in Taiwan [25]; it was later found in Italy [35], Czech Republic [48], South Korea [36], Brazil [37] and, more recently, China [49].

*Esteya* species were isolated from very different organisms, such as nematodes, insects and trees (Table 1). *Esteya vermicola* was reported to be vectored by the European oak bark beetle *Scolytus intricatus* [48] and *Oxoplatypus quadridentatus* [57]. On the other hand, *E. floridanum* was isolated from the ambrosia beetle *Myoplatypus flavicornis* [26]. An unclassified *Esteya* sp. was also isolated from ambrosia beetles *Xylosandrus crassiusculus*, *Xyleborus volvulus* and *Xyleborinus saxesenii* in Florida [58] and another unclassified *Esteya* sp. was obtained in Mexico from *Monarthrum conversum* [59], suggesting that beetles are common hosts of these fungi. Interestingly, *E. vermicola* can metabolize ethanol, a common trait in ambrosia beetle fungal mutualists, which could indicate that the fungus has coevolved with beetles [51]. In fact, ambrosia beetles have obligate associations with filamentous fungi [60,61] and this symbiosis provides the fungi with a consistent means of dispersal [62]. In the context of PWD, *Esteya* spp. were never found to be associated with *Monochamus*. Wang et al. [36] found that *E. vermicola* CNU 120806 was unable to attach or infect beetle larvae of *M. alternatus*.

The nematode host range of *E. vermicola* is quite vast. Apart from *B. xylophilus*, *E. vermicola* can parasitize *Aphelenchoides besseyi*, *A. fragariae*, *A. ritzemabosi*, *B. cocophilus*, *B. eremus*, *B. mucronatus*, *B. rainulfi*, *Ditylenchus angustus*, *D. destructor*, *D. dipsaci* and *Panagrellus redivivus*. The fungus, however, is unable to attach to and parasitize *Heterodera avenae*, *Meloidogyne incognita* and *Pratylenchus penetrans* [31,36,37,48].

## 3. Parasitism of *Esteya* spp. on the PWN

The discovery of this PWN endoparasite occurred unexpectedly, upon a survey in Taiwan, in order to prove that *B. xylophilus* was responsible for the death of exotic pine species [25]. After extracting nematodes from wood samples and a surface disinfection, the team attempted to culture them on a sterile mycelium on PDA. They observed a significant reduction in population density after 2–4 weeks and PWN cadavers revealed the presence of an endoparasitic hyphomycete [25].

The ability to lure motile nematode stages towards virulent spores is an important factor in successful parasitism [63]. *Esteya vermicola* is capable of mimicking the scent of volatile organic compounds produced by pine trees, such as α-pinene, β-pinene and camphor, to entice PWNs for nutrients, attracting them and initiating the infective cycle [64]. However, endosymbiotic bacteria are hypothesized to be responsible for the production of these volatiles [50,56] and phylogenetic analyses show that these endosymbionts are closely-related to *P. stutzeri* and affiliated with the class *Gammaproteobacteria*, being vertically transmitted from one generation to the next by sporulation [56]. Nevertheless, *E. vermicola* can attract PWNs from plant tissues, infected pine seedlings and even dead trees [65]. A PWN chemotaxis assay between the nematode-trapping fungus *Arthrobotrys brochopaga*, *Botrytis cinerea* and *E. vermicola* revealed that the latter exerted a significantly stronger attraction effect on *B. xylophilus* [66].

The infective cycle of *E. vermicola* starts when the fungus attracts the nematodes, enticing them to come into contact with the hyphae, and lunate spores adhere to the cuticle (Figure 2). These conidia usually germinate within 18–24 h, causing death when organs and tissues are completely destroyed by a mass of hyphae, growing outward and producing more lunate conidia to begin the cycle anew [67]. *Esteya floridanum* has a similar infection process, but only lunate conidia were observed to kill the nematode and germinate from its cadaver [26]. Lunate conidia are randomly attached throughout the nematode body, but the preferred sites appear to be the cephalic and tail regions [25], most likely due to the active movements of the head and tail, as well as the presence of chemoreceptors on the anterior part of the body [66]. Lunate spores from type strain ATCC 74485 adhered to the nematode’s cuticle, whereas bacilloid conidia from this isolate were not found to infect PWNs [25]. The same was assumed for most isolates of *E. vermicola*, until Wang et al. [67] found bacilloid conidia in the pseudocoelom of PWNs. Moreover, *E. vermicola* CNU 120806 blastospores proved to be equally effective as lunate conidia in suppressing PWNs [29,68]. When comparing growth rate, sporulation and infectivity of four isolates of *E. vermicola* from two different continents, there seems to be a trade-off in Asian strains ATCC 74485 and CNU 120806 between growth rate and infectivity, while European isolates CBS 115803 and CBS 100821 grow slower but produce more lunate conidia and exhibit higher virulence towards PWNs, with the Czech strain CBS 115803 being the most promising out of the four [69].

Liou et al. [25] reported that 90% percent of inoculated PWNs were found to be infected by *E. vermicola* type strain within 24 h, and almost 100% of the population was eliminated 8–10 days after inoculation. Wang et al. [36] compared the adhesion rate of *E. vermicola* to PWNs on different culture media and found that 90% of spores adhered to nematodes after 24 h on PDA, while 30% were attached to nematode cuticles on corn meal agar and only 20% on water agar. However, PWN mortality rates exceeded 90% 10 days after inoculation on all the tested media. Wang et al. [35] compared the adhesion rates and mortality rates of two European *E. vermicola* isolates: CBS 115803 fared much better at suppressing PWNs, with 100% of spores attached after 12 h and 100% nematodes dead after just 4 days, while CBS 100821 featured 93% spores attached at 24 h and reached a comparable mortality rate of 100% after 10 days. In natural conditions, Wang et al. [28] assessed the ability of the fungus to suppress the PWN in *P. densiflora* logs and observed that nematode density decreased more than 50% in logs protected by *E. vermicola* CNU 120806 at the highest conidia concentration (3 × 10^8^), compared to those inoculated with *B. cinerea*, where the nematode density increased more than 200%.

Recent experiments with green fluorescent protein (GFP)-tagged *E. vermicola* CNU 120806 revealed that this strain is capable of germinating in and colonizing pine xylem, providing evidence for the fungus’ ability to infect and kill PWNs in vivo [67,70]. The GFP-tagged *E. vermicola* made it possible to better understand the infection stages: once lunate conidia adhere to PWNs, they germinate, penetrate the cuticle and induce the formation of an infection bulb in the nematode’s pseudocoelom, away from the attachment site and without hyphae connecting the two fungal structures. Instead, an independent trophic hypha grows from the infection bulb, producing bacilloid conidia within the pseudocoelom, leading to a tangled mass of hyphae, reducing PWN motility and completely destroying organs and tissues from the inside out [67]. This suggests that *E. vermicola* can be further dispersed by infected nematodes as they migrate within their host, attracting more PWNs and initiating new infection cycles elsewhere, which are usually completed within 4 days [67,68,70]. *Esteya floridanum*, however, only seems to be capable of delaying tree death, as reported by Li et al. [26], who noticed that larch (*Larix olgensis*) and Korean pines (*P. koraiensis*) died a few weeks later compared to positive controls, when inoculated with the fungus prior to PWN infection. Nevertheless, the fungus was only tested on pine seedlings, which are usually more susceptible to pathogens and pests [71]. *Esteya floridanum* is an endophyte of pine trees [26].

De novo assembly yielded a genome of 34.2 Mb in length, with average 68.59× coverage depth. Combined de novo and homology prediction produced 8427 protein-coding genes. Further annotation results of *E. vermicola* genome revealed that there were no α- and β-pinene monoterpenoid synthase-encoding genes, suggesting that endosymbiotic bacteria might be responsible for the production of monoterpenes that attract PWNs [50,56]. Insights into the genome of *E. vermicola* revealed highly expanded endo-β-glucanase gene families, which were identified as parasitism genes in plant-parasitic nematode [72]. These enzymes likely play a crucial role in their pathogenicity, by degrading glucans from nematodes they infect [50]. Similar enzymes are also found in PWNs and since β-1,3-glucan is the main structural component of fungal cell walls, it seems probable that β-1,3-glucanases may play an important role in their mycophagous phase [73].

## 4. Application of *Esteya* spp. for a Sustainable PWD Management

### 4.1. Culturing

Nutrients exert an influence on the growth and sporulation rates of different strains of *E. vermicola*. Most isolates grow faster on nutrient-rich media. Likewise, the proportion of lunate and bacilloid conidia are linked to the availability of nutrients in the medium. CNU 120806 produces more lunate conidia in nutrient-rich media [48], whereas a negative correlation exists between adhesive conidia from ATCC 74485 and NKF 13222 and nutrients [25,36,37]. The most efficient medium to multiply blastospores of *E. vermicola* CBS 115803 was shown to be PDB, supplemented with wheat bran and pine powder [74]. When nitrogen sources (glycine and ι-leucine) were added to PDA as supplements, the fungus’ growth rate increased [75]. These two nitrogen sources also stimulated the production of lunate spores, exerting a significant influence on the adhesive and mortality rates against *B. xylophilus* and enhancing the survival rate of the fungus during abiotic stress [75]. Some mineral salts produce similar effects: CaCl_2_ and CaCO_3_ were effective in enhancing growth rate, sporulation and virulence of *E. vermicola* [76]. Carbon sources also seem to play a crucial role in the mechanisms of *Esteya* parasitism, which are yet to be clearly understood. For instance, easily metabolized carbon sources, such as glucose, were shown to result in the repression of various fungal genes needed to metabolize other carbon sources in the nematophagous fungus *Pochonia chlamydosporia* [77]. On the other hand, *E. vermicola* features more carbon transporters compared to many nematophagous fungi [50]. Remarkably, the proportion of expanded gene families of transporters in *E. vermicola* is much higher than that of other analyzed species of nematophagous fungi [50]. The transporters of expanded gene families in *E. vermicola* occurred mainly for carbon source, nitrogen source, iron and vitamin transport, which provide the basis for effectively absorbing nutrients from nematodes [50].

The optimal growth conditions for *Esteya* spp. are 26 °C and rich media, such as MEA and PDA. In these conditions, *E. vermicola* CNU 120806 reached about 4.0–4.5 cm in diameter after 8 days [36], CBS 115803 3.0–3.7 cm [48] and ATCC 74485 3.0–4.5 cm [25]. Wang et al. [75] reported that the same isolate grew to a diameter of 2.50 ± 0.17 cm (*p* < 0.05) over the same period and at the same temperature. By contrast, CBS 100821 grew slowly on PDA, reaching 2.7–3.0 cm in diameter, in the aforementioned conditions [35]. Colonies of the isolate CBS 115803 were 26–32 mm in diameter after being cultured on 2% MEA for 7 days at 25 °C [48]. *Esteya floridanum* reared on PDA grew an average of 2.73 ± 0.68 mm and 3.33 ± 0.33 mm per day, at 25 °C and 30 °C, respectively [26].

### 4.2. Mass Production and Maintenance

Mass production, stress tolerance, storage time and application of biocontrol agents are all important factors when considering biocontrol strategies, but they can be very challenging. *Esteya vermicola* mostly produces blastospores in liquid media and aerial conidia (lunate and bacilloid) on solid media [31]. Thus, blastospores can easily be multiplied in liquid medium [74], while a two-phase culture method is needed for aerial conidia production [74,78,79]. Wang et al. [75] tested the effect of mineral salts on the growth rate, sporulation and virulence of *E. vermicola* CNU 120806 and found that CaCl_2_ not only boosted the growth rate and sporulation of the fungus, but also produced the highest adhesive rate and mortality against the PWN. Wang et al. [76] observed that spores of *E. vermicola* CNU 120806 produced in media supplemented with glycine and ι-leucine had slightly higher ultraviolet (UV) and drought resistance than spores produced without nitrogen sources. Wang et al. [80] assessed the protective properties of different compounds on the conidial resistance of *E. vermicola* CBS 115803 to environmental stressors, with a formulation of 0.2% fulvic acid and 4% skimmed milk, 5% sorbitol and 0.05% CaCl_2_ protecting against UV, drought and heat stress, respectively. Xue et al. [81] found that applying herbal extracts to PDA enhanced tolerance to abiotic stress, such as drought, heat and UV radiation, in *E. vermicola* CBS 115803, CBS 100821 and CNU 120806. Wang et al. [74] noted that blastospores of *E. vermicola* CBS 115803 produced in liquid-state fermentation fared better in terms of yield and PWN mortality rate, but their germination, storage life and stress tolerance were worse when compared to aerial conidia. Xue et al. [78] showed that hydrogen peroxide promotes the sporulation of *E. vermicola* CBS 115803 conidia, under specific conditions, without them losing viability. Wang et al. [70] optimized a cost-effective formula to protect spores of *E. vermicola* CBS 115803, containing 69.9% soluble starch, 14% wheat flour, 5% PEG8000, 0.1% Span 80, 1% arabinose and 10% skimmed milk, yielding almost 80% germination rates under desiccation stress. Although lunate conidia are significantly more resistant to stress [74], Yin et al. [68] deduced that abiotic stress can be mitigated by directly injecting blastospores into pine trees. Zhu et al. [79] developed a low-cost solid-state fermentation (SSF) method for mass production of aerial conidia, using rice as a carrier to absorb the conidial suspension of *E. vermicola* CBS 115803. This study provides the basis for an economical high-quality conidia production system, without the need for additional nutrients such as carbon and nitrogen sources. While other production systems are more efficient, SSF is preferred for field application [82,83,84,85].

### 4.3. Implications for Field Use and Biocontrol Strategies

If biological control strategies are to be successful, the most efficient inoculation methods should be used for maximum effect. Wang et al. [28] observed that spraying a conidial suspension of *E. vermicola* CNU 120806 on four-year-old pine seedlings of *P. densiflora* resulted in a delay of plant death but did not prevent PWD. Nevertheless, PWN density was significantly higher in seedlings without the protective effect of *E. vermicola*. Likewise, Wang et al. [22] also reported improvements in the survival rate of *P. densiflora* seedlings against the PWN by spraying them with an *E. vermicola* CNU 120806 conidial suspension before nematode infection, which increased proportionally with colony-forming units (CFU) per mL, although the time interval between spraying pine seedlings with *E. vermicola* and PWN infection produced different effects on the survival rate: a 20 day interval between spraying and infection significantly differed from the survival rate of seedlings at 10 and 30 day intervals. In a field experiment, Wang et al. [29] inoculated 15–30 year-old *P. densiflora* with three different treatments: 2 mL of a 40% *Esteya*-infected PWN suspension (approximately 30,000 individuals, 12,000 of which were infected by the fungus), 30–40 mL of either a liquid culture or solid culture substrate of *E. vermicola* CNU 120806 and 2 mL of 40% *Esteya*-infected PWN suspension, 110 days prior to a regular PWN inoculation (1 mL, approximately 10,000 individuals), with 10 trees per treatment. In the first treatment, the tree survival rate was 80% during the months following the beginning of the experiment, decreasing to 60% within a year and stabilizing at 50% over a 3-year period. When the trees were preemptively treated with a liquid culture of *E. vermicola*, the tree survival rate was 60% during the first year, dropping to 30% the over next 12 months and remaining there for a further 4 years. On the other hand, the survival rate of pines treated with the solid substrate of *E. vermicola* started at 80%, dropping to 40% two years later and remaining there over a 4-year period [29]. The same authors also tested the effect of the fungus using a different method: in total, 10 *P. densiflora* were artificially infected with the aforementioned PWN suspension and, a week later, trees were injected with 2 mL of an 80% *Esteya*-infected PWN suspension (approximately 30,000 individuals, 24,000 of which were infected by lunate conidia of *E. vermicola*). Six trees died within 12 months, and the survival rate remained at 40% over the six-year experiment [29]. A statistical analysis showed that the protective effect of *E. vermicola* was significantly higher (*p* < 0.05) than the remedial effect of the fungus only when trees were treated with 40% infected PWN suspensions [29]. Another field experiment conducted by Yin et al. [68] demonstrated the potential of *E. vermicola*. Here, 40 healthy 15–20 year-old *P. thunbergii* were randomly divided into three groups: prevention group (15 trees pre-injected with *E. vermicola* CNU 120806 one month before PWN inoculation), cure group (15 trees injected with *E. vermicola* two and four weeks after PWN infection) and control group (10 trees inoculated with PWNs only). In the prevention group, the survival rate of these pine trees was over 70%, significantly higher than that of the cure group (40%). Wang et al. [70] carried out a field trial that provides evidence for a promising field application of *E. vermicola*. Eighty healthy 10 year-old *P. densiflora* were selected and randomly divided into four groups (prevention group, cure group, positive control and negative control), with twenty trees per group. Trees in the prevention group were inoculated with 10 mL of *E. vermicola* CNU 120806 conidial suspension by trunk injection through drilled holes and, one month later, the PWN was inoculated. The trees in the cure group were infected with PWNs two weeks before the injection of *E. vermicola* conidial suspension. The trees in the positive and negative control groups were inoculated with PWNs and sterile distilled water, respectively. The survival rates of the tested trees in the prevention and cure groups were 90% and 80%, respectively, much higher than that of the positive control group (50%). Expectedly, nematode density in wilted trees from the positive control group (2024.6 ± 146.8/g wood) was significantly higher than that observed in the prevention (301.5 ± 33.5/g wood) and cure (494 ± 117.38/g wood) treatments (*p* < 0.001). An assay in controlled conditions was performed by Li et al. [26] to assess the protective effect of *E. floridanum*. Twenty seedlings of *P. koraiensis* were preemptively inoculated with 1 mL of spores suspension of *E. floridanum* (approximately 10^6^ spores) and another twenty seedlings were left untreated. Two weeks later, all 40 plants were inoculated with a PWN suspension (approximately 1,000 nematodes per seedling). All the seedlings died after 14 weeks and 75% of Korean pine seedlings wilted 2 weeks after PWN inoculation. In plants treated with *E. floridanum*, wilting symptoms and death were deferred for 2–4 weeks and 4–6 weeks, respectively. In contrast, untreated seedlings died within 8 weeks post-PWN inoculation, whereas only 35% of Korean pines previously injected with the fungus died within that timeframe.

## 5. Future Perspectives

In over twenty years, only two *Esteya* spp. were discovered and eight isolates of *E. vermicola* described, but more possibly exist. Considering their ubiquitous nature, isolated from varying substrates and often associated with ambrosia and bark beetles, bioprospecting new *Esteya* spp. or isolates should focus heavily on those insects. Moreover, the status of the genus *Esteya* remains unresolved.

The possible role of endosymbiotic bacteria, living inside *E. vermicola*, in their parasitic lifestyle should be key areas of focus. Likewise, potential endosymbionts of *E. floridanum* are still unexplored.

Technology is ever evolving and omics are fundamental tools to elucidate important mechanisms and metabolic pathways involved in the parasitism and pathogenicity of these fungi. A knowledge gap still prevails in the complex trophic interactions between *Esteya*, PWNs and *Pinus* hosts. The use of omics to characterize *Esteya* spp. is currently very limited. High-throughput sequencing technologies, such as transcriptomics of *Esteya* during its interaction with the PWN, would provide new insights into the parasitic mechanisms of existing isolates and species.

## Figures and Tables

**Figure 1 microorganisms-10-00168-f001:**
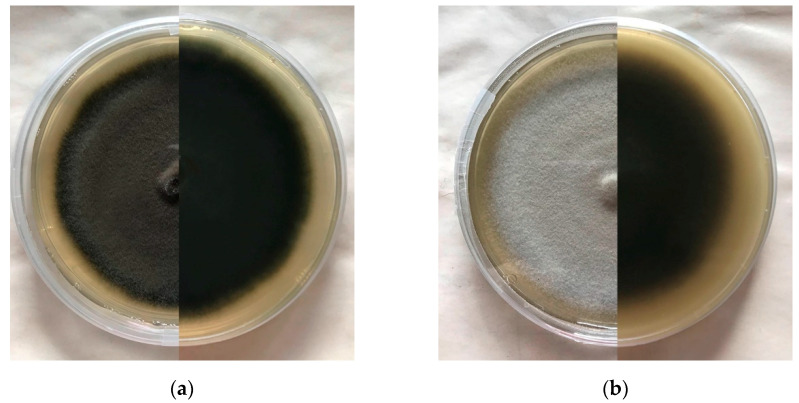
Upper and reverse perspectives of *Esteya* spp. cultured on PDA: (**a**) *E. vermicola* CNU 120806; (**b**) *E. floridanum* 14639.

**Figure 2 microorganisms-10-00168-f002:**
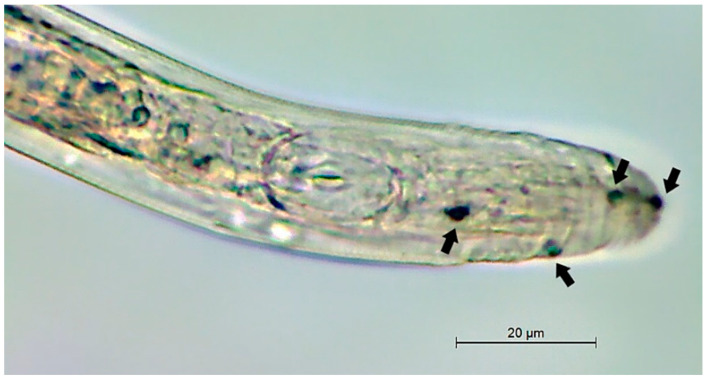
Cephalic region of *Bursaphelenchus xylophilus*, with spores of *Esteya vermicola* attached to the cuticle (arrows).

**Table 1 microorganisms-10-00168-t001:** Information on the species and isolates of *Esteya* currently described (T, type strain).

Species	Strain	Origin	Year	Substrate	Reference
*E. vermicola*	CBS 156.82	Japan	1982	Dried *Pinus*	[34,35]
	ATCC 74485^T^	Taiwan	1995	Infected PWNs from *Pinus thunbergii*	[25]
	CBS 100821	Italy	1998	Twig of *Olea europaea*	[35]
	CBS 115803	Czech Republic	1999	*Scolytus intricatus* from oak tree	[48]
	CNU 120806	South Korea	2006	Infected nematodes in forest soil	[35]
	NKF 13222	Brazil	2014	Infected *Bursaphelenchus rainulfi* from wood packing materials	[37]
	CXY 1893	China	2016	Galleries of *Tomicus yunnanensis*	[49]
	FXY 121	China	2021	*Pinus yunnanensis*	[unpublished]
*E. floridanum*	14639	United States	2017	Head of *Myoplatypus flavicornis*	[26]

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
