# Peer review of "The Potential of Esteya spp. for the Biocontrol of the Pinewood Nematode, Bursaphelenchus xylophilus"

_microorganisms, 2022, doi:10.3390/microorganisms10010168_

Round 1

Reviewer 1 Report

Very interesting manuscript describing the isolation and growth of the Esteya genus of fungus as a potential mitigant for the pinewood nematode. While I believe the microbiology section is sound, there is not enough data to demonstrate the use of Esteya fungus as a nematocide. The authors present 1 figure showing spores in the head of the nematode. I would like to see more data such as survival curves or lifespan analysis. This should be done both in a controlled lab setting and in plant infection context. 

Author Response

Dear Reviewer,

We would like to thank your great feedback on our manuscript “The potential of Esteya spp. for the biocontrol of the pinewood nematode, Bursaphelenchus xylophilus”. All suggestions were taken into account in the new version of the manuscript that we are sending for further consideration, and we address specific comments below.

While I believe the microbiology section is sound, there is not enough data to demonstrate the use of Esteyafungus as a nematicide.” – Thanks for bringing this to our attention. For more clarity, we added a new paragraph to section 3 (lines 212–226), to better highlight the ability of these fungi to kill and suppress the pinewood nematode.

I would like to see more data such as survival curves or lifespan analysis. This should be done both in a controlled lab setting and in plant infection context.” – Thank you so much for pointing out this flaw in our review; this information was indeed missing, so we added the subsection 4.3. (lines 327–387), “Implications for field use and biocontrol strategies”, where we go through studies dealing with survival rates of pine trees inoculated with the pinewood nematode and treated with Esteya spp., both in controlled conditions and field experiments.

Reviewer 2 Report

The paper is a interesting and necessary review on a promising agent of control such as the fungus Esteya spp. Of special interest, in fact, is the possibility to control one of the most devastating pest for the worldwide pine forest as is Bursaphelenchus xylophilus. 

The paper is well written and set up the methodology and references able to be used by other researcher. We can consider this paper as seminal to open new lines of research in other laboratories (and on others plant parasitic nematodes phylogenetically related, why not?), owing to the upgrade of this so specialized topic including (as the authors did) the incorporation of Esteya symbiotic bacteria.

My recomendation is to publish the paper in the present form

Author Response

Dear Reviewer,

We would like to thank your great feedback on our manuscript “The potential of Esteya spp. for the biocontrol of the pinewood nematode, Bursaphelenchus xylophilus”. 

Thank you very much for the enthusiastic response and invaluable feedback on our review.

Round 2

Reviewer 1 Report

-